# Seroprevalence of Anti-SARS-CoV-2 IgG Antibodies in Tyrol, Austria: Updated Analysis Involving 22,607 Blood Donors Covering the Period October 2021 to April 2022

**DOI:** 10.3390/v14091877

**Published:** 2022-08-25

**Authors:** Lisa Seekircher, Anita Siller, Manfred Astl, Lena Tschiderer, Gregor A. Wachter, Bernhard Pfeifer, Andreas Huber, Manfred Gaber, Harald Schennach, Peter Willeit

**Affiliations:** 1Clinical Epidemiology Team, Medical University of Innsbruck, 6020 Innsbruck, Austria; 2Central Institute for Blood Transfusion and Immunology, Tirol Kliniken GmbH, 6020 Innsbruck, Austria; 3Julius Center for Health Sciences and Primary Care, University Medical Center Utrecht, 3584 CX Utrecht, The Netherlands; 4Tyrolean Federal Institute for Integrated Care, Tirol Kliniken GmbH, 6020 Innsbruck, Austria; 5Division for Healthcare Network and Telehealth, UMIT-Private University for Health Sciences, Medical Informatics and Technology GmbH, 6060 Hall, Austria; 6Blood Donor Service Tyrol of the Austrian Red Cross, 6063 Rum, Austria; 7Department of Public Health and Primary Care, University of Cambridge, Cambridge CB1 8RN, UK

**Keywords:** SARS-CoV-2, seroprevalence, anti-spike IgG antibodies, blood donors

## Abstract

Because a large proportion of the Austrian population has been infected with SARS-CoV-2 during high incidence periods in winter 2021/2022, up-to-date estimates of seroprevalence of anti-SARS-CoV-2 antibodies are required to inform upcoming public health policies. We quantified anti-Spike IgG antibody levels in 22,607 individuals that donated blood between October 2021 and April 2022 across Tyrol, Austria (participation rate: 96.0%). Median age of participants was 45.3 years (IQR: 30.9–55.1); 41.9% were female. From October 2021 to April 2022, seropositivity increased from 84.9% (95% CI: 83.8–86.0%) to 95.8% (94.9–96.4%), and the geometric mean anti-Spike IgG levels among seropositive participants increased from 283 (95% CI: 271–296) to 1437 (1360–1518) BAU/mL. The percentages of participants in categories with undetectable levels and detectable levels at <500, 500–<1000, 1000–<2000, 2000–<3000, and ≥3000 BAU/mL were 15%, 54%, 15%, 10%, 3%, and 3% in October 2021 vs. 4%, 18%, 17%, 18%, 11%, and 32% in April 2022. Of 2711 participants that had repeat measurements taken a median 4.2 months apart, 61.8% moved to a higher, 13.9% to a lower, and 24.4% remained in the same category. Among seropositive participants, antibody levels were 16.8-fold in vaccinated individuals compared to unvaccinated individuals (95% CI: 14.2–19.9; *p*-value < 0.001). In conclusion, anti-SARS-CoV-2 seroprevalence in terms of seropositivity and average antibody levels has increased markedly during the winter 2021/2022 SARS-CoV-2 waves in Tyrol, Austria.

## 1. Introduction

The coronavirus disease 2019 (COVID-19) pandemic situation remains volatile due to the ongoing emergence of new variants of concern and because specific population subgroups remain vulnerable to severe acute respiratory syndrome coronavirus type 2 (SARS-CoV-2) infection. High levels of SARS-CoV-2 antibodies are associated with a lower infection risk and severe COVID-19 [1,2,3,4,5,6]. However, the level of protection is not assumed to be identical for all viral variants [7,8,9,10], and multiple factors contribute to the degree of individual immune response, such as age [11,12], sex [11,12], the variant which caused an infection [13,14], the severity of infection [15,16], vaccine products [17,18,19], and the interdose interval [20,21], among others. Furthermore, protection against SARS-CoV-2 infection diminishes over time as anti-SARS-CoV-2 antibodies wane [10,11,22]. Therefore, continuous monitoring of the seroprevalence of antibodies is important to help appraise the latest immune status in the population.

Since June 2020, our study team has been assessing continuously the seroprevalence of anti-SARS-CoV-2 antibodies in blood donors from the Federal State of Tyrol, Austria. Our two prior reports [23,24] showed that seroprevalence increased to 11.6% by the end of 2020, then further increased sharply to 68.3% by mid-2021 coinciding with vaccination rollout, before it then plateaued at 82–83% in August and September 2021. In the end of December 2020, SARS-CoV-2 vaccination initiatives were rolled out in the Federal State of Tyrol accompanied by the implementation of booster vaccinations lasting from October 2021 onward. By the end of April 2022, 75.0% and 72.2% of the Tyrolean population had received one and two doses of any SARS-CoV-2 vaccine, respectively, and 52.9% were vaccinated with a booster dose (Appendix A) [25]. Since our latest report, Tyrol has also experienced waves of infections with the Delta and the Omicron variants. Between October 2021 and April 2022, these waves have cumulatively led to 310,308 cases of infection, corresponding to an estimated 35–40% of the general population (Appendix A) [26,27]. Furthermore, infection with the Omicron variant is more commonly asymptomatic or paucisymptomatic [28,29,30,31,32], and therefore it is plausible that cases of Omicron infection more commonly remain undetected than infections with earlier variants, which suggests that the “true” number of cases is probably even higher. The impact of the high number of infections during winter 2021/2022 on the current seroepidemiological status of the population is likely to be substantial but remains to be determined reliably in an up-to-date seroepidemiological study.

To provide clarity, we analyzed most recent data from our ongoing study on blood donors from the Federal State of Tyrol, Austria, covering the period between October 2021 and April 2022. In this report, we set out to quantify (i) the anti-Spike IgG seropositivity for each month, (ii) the levels of anti-Spike IgG antibodies, and (iii) differences in levels by region, age, sex, and vaccination status.

## 2. Materials and Methods

Details on the design of this ongoing study were previously described [23,24]. Briefly, individuals were eligible for inclusion if they (i) were aged between 18 and 70 years; (ii) were permanent residents in Tyrol; (iii) and fulfilled the general requirements for donating blood, including being in a healthy state. In the present analysis, we included individuals who donated blood between 1 October 2021 and 28 April 2022 at 119 blood donation locations spread across all districts of Tyrol. Of 23,552 individuals eligible for inclusion, 22,607 participated, corresponding to a participation rate of 96.0%. Data on age, sex, and SARS-CoV-2 vaccination were collected routinely as part of every blood donation.

Serum samples were drawn, cooled at 4 °C, and normally processed within 30 h at the laboratory of the Central Institute for Blood Transfusion and Immunology of the University Hospital in Innsbruck, Austria. As in the prior rounds of this ongoing study, we assessed samples for anti-SARS-CoV-2 IgG antibodies targeting the receptor-binding domain of the spike protein (anti-S IgG) using the Abbott SARS-CoV-2 IgG II chemiluminescent microparticle immunoassay analyzed on the Alinity i instrument (Abbott Ireland, Sligo, Ireland). As recommended by the assay manufacturer, we classified anti-S IgG values ≥ 7.1 Binding Antibody Units per milliliter (BAU/mL) as seropositive and <7.1 BAU/mL as seronegative. This cut-off value has a sensitivity of 99.35% (95% confidence interval (CI): 96.44–99.97%) at ≥15 days after COVID-19 onset (postsymptom onset) and a specificity of 99.60% (99.22–99.80%).

We quantified seropositivity of anti-S IgG antibodies with Agresti-Coull 95% CIs [33] for each month between October 2021 and April 2022. We performed sensitivity analyses on the monthly seropositivity that (i) restricted the analysis to individuals with repeat donations since July 2017 to limit the possibility of selection bias related to offering free antibody tests as part of our study and (ii) applied an age and sex standardization using the structure of the total population in Tyrol aged 18–70 years [27] to examine generalizability of the study estimates.

Among seropositive participants, we summarized the level of anti-S IgG antibodies as geometric means and 95% CIs. We estimated the difference in anti-S IgG levels across months using a linear mixed model with a random intercept at the participant level. Additionally, we calculated the corresponding percentage across different categories of anti-S IgG antibody levels (undetectable levels and detectable levels at <500, 500–<1000, 1000–<2000, 2000–<3000, and ≥3000 BAU/mL). These categories were selected based on the results of the phase-4 open-label Shieldvacc-2 study, a clinical trial conducted in Tyrol that investigated the association between anti-S IgG antibody levels and the risk of breakthrough infection among individuals vaccinated with two doses of BNT162b2 and showed that the level of protection is dependent on the level of antibodies [1]. Furthermore, among participants with repeat measurements, we quantified the shift of participants across these categories of anti-S IgG antibody levels using the first and last available measurement of anti-S IgG levels between October 2021 and April 2022.

For the most recent measurements in April 2022, we investigated differences in seropositivity by district, age groups, sex, and vaccination status using χ^2^-tests and—among seropositive participants—differences in anti-S IgG antibody levels using linear regressions. For the latter analysis, we log-transformed antibody values and back-transformed results to quantify percentage differences. *p*-values ≤ 0.05 were deemed as statistically significant, and all statistical tests were two-sided. Analyses were carried out with Stata 15.1 and R 4.1.0.

## 3. Results

The results are reported in accordance with the Strengthening the Reporting of Observational Studies in Epidemiology (STROBE) guidelines (Appendix A).

### 3.1. Study Population

Table 1 summarizes the characteristics of the 22,607 participants enrolled in our study. At the first blood donation during the course of our study (“baseline”), the median age of participants was 45.3 years (interquartile range (IQR): 30.9–55.1), 41.9% were female, and 84.9% were vaccinated against SARS-CoV-2. In total, 25,561 anti-S IgG antibody tests were performed; 2711 participants had two or more measurements over a median follow-up duration of 4.2 months (IQR: 3.0–5.3).

### 3.2. Seroprevalence of Anti-S IgG Antibodies by Time

Table 2 shows anti-S IgG seroprevalence analyzed cross-sectionally for each month between October 2021 and April 2022. Seropositivity was 84.9% (95% CI: 83.8–86.0%) in October 2021 and increased to 95.8% (94.9–96.4%) in April 2022. In sensitivity analyses (Appendix A), seropositivity estimates were robust when standardized to the age and sex distribution in Tyrol, e.g., 85.3% (84.2–86.4%) in October 2021 and 95.9% (95.1–96.6%) in April 2022. Furthermore, results were similar when restricting the analysis to participants who had already donated blood before study enrolment, e.g., 87.0% (85.8–88.1%) in October 2021 and 95.7% (94.8–96.5%) in April 2022.

### 3.3. Anti-S IgG Antibody Levels

We observed an increase in anti-S IgG antibody levels during the course of our study. As shown in Table 2, among seropositive participants, the geometric mean antibody level rose from 283 BAU/mL in October 2021 (95% CI: 271–296) to 1437 BAU/mL in April 2022 (1360–1518), corresponding to a 5.3-fold change (95% CI: 5.0–5.7) as estimated from a linear mixed model. Furthermore, we inspected the percentages of individuals across pre-specified categories of anti-S IgG antibody levels for each month during the course of our study (Figure 1). The percentages of participants with undetectable levels and detectable levels at <500, 500–<1000, 1000–<2000, 2000–<3000, and ≥3000 BAU/mL were 15%, 54%, 15%, 10%, 3%, and 3% in October 2021 vs. 4%, 18%, 17%, 18%, 11%, and 32% in April 2022.

In addition, we performed a longitudinal analysis of the 2711 participants with repeat measurements of antibody levels (12% of the total study population). Figure 2 depicts the shift of these participants across categories of anti-S IgG antibody levels between the first and last available follow-up measurement. The upper part of the matrix in Figure 2 shows that 61.8% moved to a higher category, the lower part that 13.9% moved to a lower category, and the diagonal that 24.4% remained the same category. Separate results for vaccinated and unvaccinated participants are provided in Appendix A. Notably, only few participants who remained unvaccinated throughout the duration of the follow-up moved to categories with anti-S IgG antibody levels higher than 500 BAU/mL (25 out of 322, i.e., 7.8%).

### 3.4. Cross-Sectional Correlates of Anti-S IgG Seroprevalence and Antibody Levels

Table 3 summarizes differences by district, age groups, sex, and vaccination status and—among seropositive participants—differences in anti-S IgG antibody levels. Seropositivity in April 2022 was similar across the nine districts of Tyrol (*p*-value = 0.957) and in females and males (*p*-value = 0.463). However, seropositivity differed significantly across age groups (*p*-value < 0.001). Furthermore, seropositivity was significantly lower in unvaccinated (66.6%; 61.3–71.5%) than in vaccinated individuals (99.7%; 99.3–99.8%) (*p*-value < 0.001).

Among seropositive participants, antibody levels were 16.8-fold (95% CI: 14.2–19.9; *p*-value < 0.001) in vaccinated individuals (geometric mean: 1813 (95% CI: 1731–1899)) compared to unvaccinated individuals (geometric mean 108 (86–136)). Additionally, antibody levels differed by age group, e.g., they were 34% lower in participants aged ≥65 years (95% CI: −52 to −8%) compared to participants aged <25 years. We also detected differences between individual districts, e.g., antibody levels were 27% lower in individuals from Kitzbühel (−40 to −11%) than in individuals from Schwaz. We found no significant differences by sex (*p*-value = 0.726).

## 4. Discussion

The present study reports the anti-S IgG-specific seroprevalence and antibody levels in 22,607 Tyrolean blood donors following waves of infections with Delta and Omicron variants during winter 2021/2022. We observed a high seropositivity with 95.8% in April 2022 and revealed differences in seropositivity by age groups and vaccination status but not by sex and district. Furthermore, anti-S IgG antibody levels increased significantly during the course of our study. Among seropositive participants, antibody levels were higher in individuals aged < 25 years compared to older age groups and in vaccinated individuals compared to unvaccinated. We also detected differences in antibody levels across districts in Tyrol.

Our study provides novel insights in three specific areas. First, the current analysis delivers up-to-date and much needed population-based seroprevalence estimates after a sizeable proportion of the Austrian population has been infected with SARS-CoV-2 during winter 2021/2022. Furthermore, prior evidence from seroepidemiological studies in Austria is restricted to the period from April 2020 to December 2021 [23,24,34,35,36,37,38,39,40,41,42,43]. Our latest report of Tyrolean blood donors showed that 82.7% had detectable anti-S IgG antibodies in September 2021 [24]. In the present report, we were able to extend our study period for seven additional months, and we have revealed that seroprevalence further increased to 95.8% by April 2022, translating to one out of 25 individuals being immuno-naive.

Second, our study shows a major increase in average anti-S IgG antibody levels in this blood donor cohort. In specific, the geometric mean level increased from 283 to 1437 BAU/mL during the course of our study. When we categorized participants according to their antibody levels, we observed that the proportion of people in categories with higher levels increased substantially by the end of the study. This result was further substantiated by a longitudinal analysis of individuals that donated blood repeatedly, in which observed a major shift of individuals to higher antibody categories. The increase in anti-S IgG antibodies is predominantly attributable to high infection rates (i.e., a total of 310,308 cases in Tyrol between October 2021 and April 2022 [26]) and the rollout of booster vaccinations. Appendix A depicts the vaccine coverage in the total population of Tyrol stratified by vaccine dose. The booster vaccinations increased from 25.7% at the beginning of November 2021 to 49.2% at the end of January 2022 but stagnated afterward [25]. Furthermore, we observed a reduction of antibody levels in some individuals. This is in line with previous studies that demonstrated significant waning of antibody levels over time [11,22]. Approximately one in four participants had similar antibody levels at baseline and follow-up. Of those, a large proportion was seronegative or had anti-S IgG antibody levels < 500 BAU/mL. Persisting low antibody levels were mainly determined by lack of vaccination (Appendix A).

Performing quantitative analyses of anti-S IgG antibody levels rather than solely focusing on the presence vs. absence of antibodies (i.e., seropositivity) is important because several studies have shown that levels are inversely associated with the risk of acquiring SARS-CoV-2 infection [1,2,4,5,6]. For example, in a study from Tyrol conducted in individuals that had received two doses of the BNT162b2 vaccine, we have recently shown that protection from breakthrough infection was more pronounced when anti-S IgG levels were higher. For instance, the hazard ratios for breakthrough infection was 0.20 (0.06–0.64) for individuals with anti-S IgG levels ≥ 3000 BAU/mL compared to individuals with anti-S levels < 500 BAU/mL [1]. While it renders plausible that our study population exhibited greater protection from SARS-CoV-2 by the end of the study period in April 2022, there is uncertainty about the level of protection that specific antibody levels confer from infection with different Omicron subvariants or other variants of concern [44]. Data indicate that anti-S IgG antibodies are, if at all, only poorly correlated with neutralizing antibodies against Omicron variants [13,14,45], and Omicron infections induce low-level SARS-CoV-2-neutralizing activity [13,14,30,46]. Thus, despite the fact that immunity obtained from vaccination or infection against Omicron BA.1 and BA.2 may be high, the new Omicron subvariants BA.4 and BA.5 lead to surges of infections [46,47].

Third, our study investigated differences in seropositivity and antibody levels across prespecified participant characteristics including the most recent measurements from April 2022. We observed no regional differences in seropositivity across the nine districts of the Tyrol. Contrarily, we found district-specific differences in our previous study, in which we detected the lowest seroconversion in the district of Lienz with 74.3% and the highest in the district of Schwaz with 87.6% in the period between July and September 2021 [24]. In comparison, in the current analysis, seropositivity was 96.2% in Lienz and 95.6% in Schwaz in April 2022. However, anti-S IgG antibody levels differed across individual districts, which is likely due to differing uptake rates of the first booster dose across the districts. Additionally, as in our previous study [24], higher antibody levels were visible in individuals aged < 25 years compared to older age groups. Humoral immune responses following vaccination against SARS-CoV-2 have been reported to be age-dependent and to be more robust in younger individuals [11,48]. Another potential explanation is that the average time span since the latest vaccination is likely to be shorter in younger individuals as a consequence the age-staggered rollout of the vaccination program. Furthermore, while in our study anti-S IgG antibodies were detectable in almost all vaccinated individuals (99.7%), approximately two-thirds of unvaccinated individuals were seropositive. Earlier estimates of seropositivity in Austria among unvaccinated individuals from 2021 ranged from 21.7% [43] to 26.1% [24].

In addition, in line with our previous findings, anti-S IgG antibody levels were 16.8-fold in vaccinated compared to unvaccinated participants. The reason for this remarkable difference is not entirely clear, and there may be various explanations. First, an infection with the Omicron subvariants may induce lower antibody levels than vaccination. Several studies reported lower humoral immune responses after an infection with Omicron compared to other variants. This could be associated with milder symptoms caused by Omicron infections [28,29,30,31,32], since higher antibody levels and its persistence are correlated with disease severity in COVID-19 [15,16]. Recently, a study observed a lower humoral immune response after infection in Omicron convalescent individuals compared to individuals with two-dose mRNA vaccinations [14]. Second, the hybrid immunity resulting from a combination of SARS-CoV-2 exposures of vaccination and infection may be characterized by higher antibody levels than the immune response to infection alone. Given the unavailability of information on prior SARS-CoV-2 infections, we could not investigate the subgroup of participants with hybrid immunity in our study. However, since in the Austrian population the number of breakthrough infections despite immunization is rising [49], we can assume that a significant proportion of vaccinated participants also received an infection-induced booster. Overall evidence suggests that individuals with hybrid immunity may possess a more robust immune responses [30,50,51,52,53,54] and a higher protection from infection and severe COVID-19 [55,56] compared to individuals with either vaccination or previous infection alone. It is hypothesized that the protective advantage in individuals with hybrid immunity is that they generate more SARS-CoV-2 receptor binding domain-specific memory B cells and variant-neutralizing antibodies and a distinct population of interferon gamma and Interleukin 10-expressing memory SARS-CoV-2 spike-specific CD4+ T cells than previously naive individuals [57].

Our study has several strengths. First, our study population is adequately sized to quantify time-specific seropositivity and average antibody levels estimated as well as differences in these parameters by relevant population subgroups. Another strength of our study is the remarkably high participation rate of 96.0%, which we achieved because of the high interest of participants in their antibody results and the smooth integration within the routine processes at the blood donation events. Furthermore, our findings are likely to be generalizable to age groups between 18 and 70 years, since blood donors represent a healthy subgroup of the adult general population, and the age and sex structure of our study sample was similar to the Tyrolean population. In addition, the percentage of unvaccinated individuals in our study population (15.1%) was slightly lower compared to the overall population of Tyrol (19.5% aged 15 to 74 years at the median date of baseline, i.e., 13 January 2022) [27,58]. Moreover, our study period covers seven months, allowing it to reflect the seroconversion after a considerable proportion of the population has been infected by the Delta and the Omicron variants. Despite these strengths, our study also has limitations. First, we did not measure anti-SARS-CoV-2 IgG antibodies targeting the nucleocapsid protein, which hampers the distinction between vaccination-induced and infection-and-vaccine-induced seroprevalence. Second, in our large-scale study, it was unfeasible to also perform measurements of cellular immune responses or neutralizing antibodies, which would have provided a better insight into the immune status of the study population. Third, children, adolescents, individuals aged ≥ 70 years, and individuals with severe comorbidities were excluded from the analysis by design (see exclusion criteria for blood donation). It is known that older age and specific comorbidities influence the severity of COVID-19, which may also affect the level of SARS-CoV-2 antibodies. Fourth, the lack of information on prior SARS-CoV-2 infections, variants of infection, and vaccination details (dates received, number of doses, vaccine types), impedes more detailed analyses on specific subgroups.

## 5. Conclusions

In conclusion, anti-SARS-CoV-2 seroprevalence in terms of seropositivity and average antibody levels increased markedly during the winter 2021/2022 SARS-CoV-2 waves in Tyrol, Austria.

## Figures and Tables

**Figure 1 viruses-14-01877-f001:**
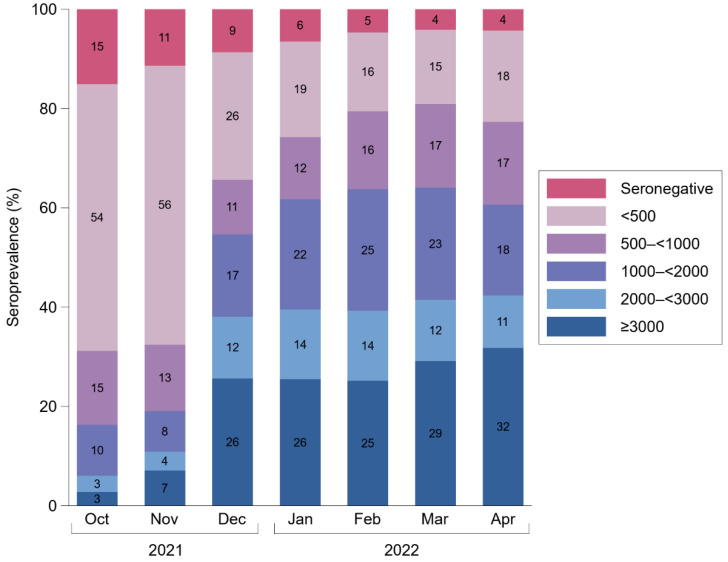
Percentage of participants in categories of anti-S IgG antibody levels in Binding Antibody Units per milliliter between October 2021 and April 2022. Seronegativity corresponds to anti-S IgG levels < 7.1 Binding Antibody Units per milliliter. The analysis involved data on 25,561 measurements taken from 22,607 individuals.

**Figure 2 viruses-14-01877-f002:**
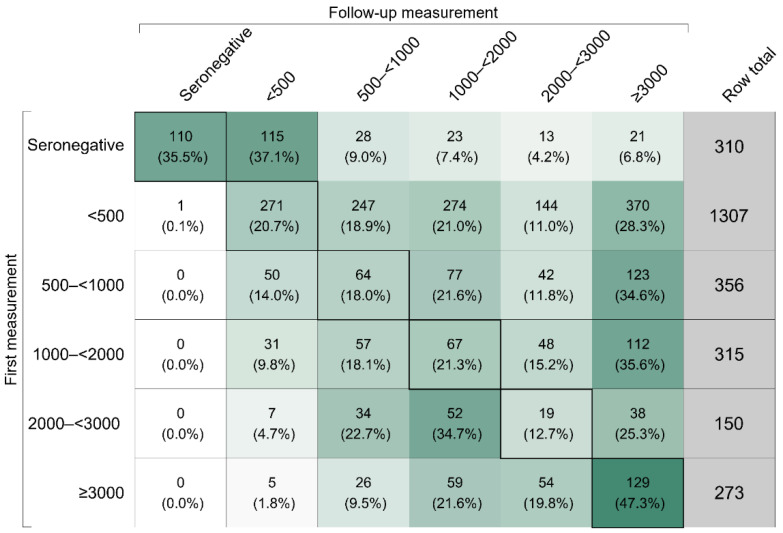
Shift of participants across categories of anti-S IgG antibody levels in Binding Antibody Units per milliliter between first and last available follow-up measurement between October 2021 and April 2022. Seronegativity corresponds to anti-S IgG levels < 7.1 Binding Antibody Units per milliliter. The intensity of the cell color reflects the row percentage of the cell. The diagonal of the matrix indicates the number (row percentage) of participants who remained in the same category of anti-S IgG antibody level between the first and last available follow-up measurement. The upper part of the matrix depicts the number (row percentage) of participants that were shifted to a higher category, i.e., they had an increase in anti-S IgG antibody levels between the first and last available follow-up measurements. The lower part of the matrix depicts the number (row percentage) of participants that were shifted to a lower category, i.e., they had a decrease in anti-S IgG antibody levels between the first and last available follow-up measurement. The analysis was restricted to participants with at least one follow-up measurement and involved data on 2711 individuals.

**Table 1 viruses-14-01877-t001:** Characteristics of participants enrolled in our study; Tyrol, Austria; October 2021–April 2022 (*n* = 22,607).

	No. of Participants	No. (%) or Median (IQR)
**Baseline data**		
Age in years—median (IQR)	22,607	45.3 (30.9–55.1)
Female sex—no. (%)	22,607	9477 (41.9%)
Vaccinated against SARS-CoV-2 *—no. (%)	22,597	19,181 (84.9%)
First donation since July 2017—no. (%)	22,607	4843 (21.4%)
**Repeat donations during study**		
Participants with ≥2 donations—no. (%)	22,607	2711 (12.0%)
Follow-up duration in months—median (IQR)	2711	4.2 (3.0–5.3)

Abbreviations: IQR, interquartile range. * Participants received at least one dose of any vaccine against SARS-CoV-2.

**Table 2 viruses-14-01877-t002:** Seroprevalence and geometric mean of anti-S IgG antibodies in Tyrolean blood donors aged 18–70 years; Tyrol, Austria; October 2021–April 2022 (total *n* = 22,607).

Month	No. of Measurements	% Seropositive (95% CI)	Geometric Mean (95% CI) in BAU/mL *
**2021**			
October	4230	84.9 (83.8–86.0)	283 (271–296)
November	3833	88.7 (87.6–89.6)	313 (298–329)
December	3363	91.3 (90.3–92.3)	1036 (981–1094)
**2022**			
January	4567	93.5 (92.8–94.2)	1215 (1163–1268)
February	3059	95.3 (94.5–96.0)	1354 (1291–1421)
March	3753	95.9 (95.2–96.5)	1489 (1426–1555)
April	2756	95.8 (94.9–96.4)	1437 (1360–1518)

Abbreviations: BAU/mL, Binding Antibody Units per milliliter; CI, confidence interval. * Among seropositive participants.

**Table 3 viruses-14-01877-t003:** Cross-sectional correlates of seroprevalence and anti-S IgG antibody levels; Tyrol, Austria; April 2022.

	Seropositivity (*n* = 2753)		Anti-S IgG Antibody Levels * (*n* = 2636)			
	N	% Seropositive (95% CI)	N	Geometric Mean (95% CI) in BAU/mL	% Difference (95% CI) vs. Reference	*p*-Value
**Age groups (years)**						
<25	342	98.8 (96.9–99.7)	338	2013 (1769–2290)	(Reference)	
25–< 35	480	95.2 (92.9–96.8)	457	1341 (1174–1531)	−33 (−46 to −18)	<0.001
35–< 45	508	94.3 (91.9–96.0)	479	1188 (1036–1362)	−41 (−52 to −28)	<0.001
45–< 55	707	93.9 (91.9–95.5)	664	1444 (1294–1611)	−28 (−41 to −13)	<0.001
55–< 65	617	97.7 (96.2–98.7)	603	1469 (1307–1651)	−27 (−40 to −12)	0.001
≥65	99	96.0 (89.7–98.7)	95	1338 (975–1837)	−34 (−52 to −8)	0.014
**Sex**						
Female	1125	96.1 (94.8–97.1)	1081	1421 (1302–1550)	(Reference)	
Male	1628	95.5 (94.4–96.4)	1555	1449 (1350–1556)	+2 (−9 to +14)	0.726
**SARS-CoV-2 vaccination**						
No	325	66.6 (61.3–71.5)	216	108 (86–136)	(Reference)	
Yes	2428	99.7 (99.3–99.8)	2420	1813 (1731–1899)	+1579 (+1317 to +1888)	<0.001
**District in Tyrol**						
Schwaz	366	95.6 (93.0–97.3)	350	1710 (1461–2001)	(Reference)	
Innsbruck-Land	497	95.0 (92.6–96.6)	472	1530 (1353–1730)	−11 (−27 to 9)	0.275
Innsbruck-Stadt	193	97.4 (94.0–99.1)	188	1681 (1403–2014)	−2 (−24 to +27)	0.897
Kufstein	212	95.8 (92.0–97.9)	203	1467 (1200–1792)	−14 (−33 to +10)	0.228
Kitzbühel	596	95.8 (93.9–97.2)	571	1251 (1104–1418)	−27 (−40 to −11)	0.001
Imst	315	95.9 (93.0–97.6)	302	1339 (1135–1580)	−22 (−37 to −2)	0.031
Landeck	15	93.3 (68.2–100.0)	14	2117 (1038–4320)	+24 (−43 to +167)	0.586
Reutte	169	95.3 (90.8–97.7)	161	1401 (1129–1738)	−18 (−37 to +7)	0.147
Lienz	390	96.2 (93.7–97.7)	375	1345 (1161–1559)	−21 (−36 to −3)	0.025

Abbreviations: BAU/mL, Binding Antibody Units per milliliter; CI, confidence interval. * Among seropositive participants. In these analyses, we only included measurements from April 2022. We investigated differences by age groups, sex, vaccination status, and district in Tyrol in seropositivity using χ^2^-tests and among detectable values of anti-S IgG antibody levels using linear regression.

## Data Availability

Data on COVID-19 cases [26], on the age and sex structure of the population in Tyrol [27], on SARS-CoV-2 vaccinations stratified by vaccine dose [25], and on SARS-CoV-2 vaccinations stratified by age groups in Tyrol [58] are publicly available. Tabular data on the blood donor cohort can be requested from the corresponding authors by researchers who submit a methodologically sound proposal (including a statistical analysis plan); participant-level data on the blood donor cohort cannot be shared due to regulatory restrictions.

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
