# Peer review of "Seroprevalence of Anti-SARS-CoV-2 IgG Antibodies in Tyrol, Austria: Updated Analysis Involving 22,607 Blood Donors Covering the Period October 2021 to April 2022"

_viruses, 2022, doi:10.3390/v14091877_

Round 1
Reviewer 1 Report
The manuscript submitted by Seekircher et al. presents interesting data on seroprevalence in a large population based study. The authors describe nicely the growing antibody levels over the time of autumn/winter 2021/2022.
In my opinion, it should be made clearer, that the increase in antibody levels are mainly not based on an individual folllow-up, as only 12% are repeating donors. The results from this group, as nicely described in Fig. 2, are very important and could be discussed in more detail. Is there any detectable cause for persisting low or decreasing antibody levels?
The limitations of the study should be discussed more in detail. Also, the lack of participants with severe comorbidities should be included in the limitations, as these participants are known to have a higher risk for SARS-CoV-2.
Additional analysis, such as detection of the cellular immune response or neutralizing antibodies could improve the presented data.
Reviewer 2 Report
Seekircher et al. study provide an insight into the evolution of anti-SARS-CoV-2 IgG antibodies seroprevalence in Tyrol's population from October 2021 to April 2022. The study is well written and fairly clearly presented. Few points, comments and additional observations could further improve the manuscript nonetheless (as see below).
Major:
- Authors need to give some specific information relative to Austria and the Tyrol region vaccine rollout and boosters rollout. Every countries had different timelines and it is important for the reader to know these dates of vaccine deployment? It will help understand the data, especially the jump seen in December 2021 (Fig. 1).
- Is the population skewed? I wonder if blood donor might have overall a skewed behavior relative to health concern (perhaps more conscious of health concerns (more likely to be vaccinated and/or less likely to expose themselves)). Although unlikely to significantly impact the data and the analysis performed, it could be an interesting insight as if the representation is an under/over-estimation. Is there such data/articles existing relative to other vaccines and blood donors such as Influenza for instance?
- Figure 2 (and Fig S1) are a bit confusing and need to be explained in more details in the results section. Authors should add a couple of sentences to layout the purpose of the graph and further comment the results. How do the authors defined first and follow-up measurement? (whether the time points are October to April or 1 months apart would skew the data to some degree). Do the authors have information relative to booster shots for the individuals vaccinated?
- Fig S1B - Is it unvaccinated people only that remained unvaccinated throughout the duration of the follow-up?
Minor:
- Figure 1 : Instead of using the month numbers, authors could probably use the months' 3 letter acronyms on the x-axis.
Round 2
Reviewer 2 Report
Authors made significant effort to address my comments.